# Cooperative Classification and Rationalization for Graph Generalization

## ABSTRACT

Graph Neural Networks (GNNs) have achieved impressive results in graph classification tasks, but they struggle to generalize effectively when faced with out-of-distribution (OOD) data. Several approaches have been proposed to address this problem. Among them, one solution is to diversify training distributions in vanilla classification by modifying the data environment, yet accessing the environment information is complex. Besides, another promising approach involves rationalization, extracting invariant rationales for predictions. However, extracting rationales is difficult due to limited learning signals, resulting in less accurate rationales and diminished predictions. To address these challenges, in this paper, we propose a Cooperative Classification and Rationalization (C2R) method, consisting of the *classification* and the *rationalization* module. Specifically, we first assume that multiple environments are available in the *classification* module. Then, we introduce diverse training distributions using an environment-conditional generative network, enabling robust graph representations. Meanwhile, the *rationalization* module employs a separator to identify relevant rationale subgraphs while the remaining non-rationale subgraphs are de-correlated with labels. Next, we align graph representations from the *classification* module with rationale subgraph representations using the knowledge distillation methods, enhancing the learning signal for rationales. Finally, we infer multiple environments by gathering non-rationale representations and incorporate them into the *classification* module for cooperative learning. Extensive experimental results on both benchmarks and synthetic datasets demonstrate the effectiveness of C2R. Code is available at https://anonymous.4open.science/r/Codes-of-C2R-ECA2.

## KEYWORDS

Graph Generalization, Out-Of-Distribution, Rationalization

ACM Reference Format:
Anonymous Author(s). 2018. Cooperative Classification and Rationalization for Graph Generalization. In *Proceedings of The Web Conference (WWW '24)*. ACM, New York, NY, USA, 10 pages. https://doi.org/XXXXXXX.XXXXXXX

## 1 INTRODUCTION

Graph Neural Networks (GNNs) have showcased remarkable achievements in graph classification across various domains [13, 15, 36, 42]. However, most existing approaches assume that the training and

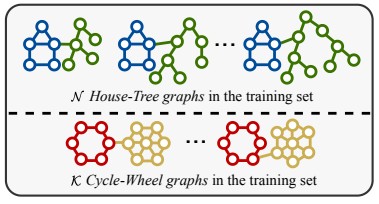 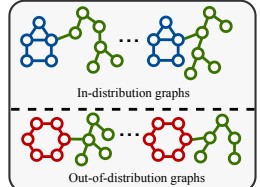

| Training dataset | Test dataset |
|---|---|

**Figure 1: An example of the motif type prediction, where the *House* and *Cycle* are motif labels, and *Tree* and *Wheel* are base subgraphs. Within the training set, there is a substantial disparity in the occurrence of *House-Tree* graphs compared to *Cycle-Wheel* graphs. This means that the number of *House-Tree* graphs ($\mathcal{N}$) greatly exceeds the number of *Cycle-Wheel* graphs ($\mathcal{K}$). Consequently, GNNs trained on such imbalanced data distributions tend to exhibit higher accuracy when handling in-distribution data, specifically *House-Tree* graphs. However, these models are more susceptible to making errors when faced with out-of-distribution (OOD) data, such as *Cycle-Tree* graphs.**

test data distributions are identical, a condition that is often challenging to meet in real-world scenarios. In reality, the test set distribution tends to differ significantly from that of the training set, posing difficulties for GNNs to generalize effectively when facing out-of-distribution (OOD) data [2, 10, 12, 22, 25].

Considering Figure 1, in the motif types prediction, we yield the motif types on a graph comprising motifs (e.g., *Cycle* and *House*) and base subgraphs (e.g., *Tree* and *Wheel*). The training dataset exhibits a prominent occurrence of *House*-motifs in conjunction with *Tree* base subgraphs, constituting a significant proportion of the dataset. In contrast, other types of data are infrequently observed, potentially inducing an overreliance of GNNs on the statistical association between *House* and *Tree* to achieve high prediction accuracy. Therefore, in the in-distribution test set, GNNs can accurately identify the *House-Tree* data as *House*. However, this dependence on bias may lead to errors when confronted with OOD data. For example, when faced with *Cycle-Tree* data, GNNs may misclassify the motifs as another category, such as *House*.

To tackle the OOD generalization challenge in graph classification, numerous approaches have been proposed. Among them, one intuitive approach [29, 34, 35] is in the process of vanilla classification to introduce greater diversity in the training distributions by modifying the data environment of the training set [6]. Specifically, we can manipulate the training data under environments that are de-correlated with the true labels. However, accessing and observing information about the environment is typically complex, rendering this approach currently impractical.

Besides, another recent promising category of methods is rooted in the concept of rationalization [9, 23, 32, 37]. These methods first

identify the rationales (aka, explanations or evidences) in the graph that are relevant to the labels and subsequently make predictions solely based on the extracted rationales. Simultaneously, subgraphs not identified as rationales are treated as environments, enabling the creation of counterfactual samples to enhance the model's generalization capability. The key challenge in this approach lies in accurately extracting the rationales. However, since the learning signals for the rationales solely derive from the comparison between the prediction results of the rationalization method and the true labels, there exists a vast exploration space to find the correct rationales. Consequently, the model may struggle to converge to the optimal rationales and predictions, leading to less accurate rationale generation [31, 43]. As a result, some nodes that should be part of the rationales may be incorrectly predicted as non-rationale nodes, ultimately diminishing the final prediction performance.

To address the limitations of the aforementioned approaches and further tackle the OOD generalization problem in graph classification, we propose a **C**ooperative **C**lassification and **R**ationalization (**C2R**) method for graph generalization. Specifically, our methodology comprises two key components, including the *classification* and *rationalization* modules. In the *classification* module, we first assume that multiple environments are available, with each sample associated with a specific environment. To enhance the diversity of the data distribution, we employ an *environment-conditional generator* that maps samples from the current environment to other environments, composing new counterfactual samples. Notably, as the environment does not influence task predictions, the labels of the generated samples remain unaltered. Finally, by amalgamating the original and generated samples during model training, we are able to derive graph representations characterized by robust generalization capabilities.

Simultaneously, within the *rationalization* module, we employ a separator to identify and extract subsets of rationale subgraphs. The remaining non-rationale subgraphs are de-correlated with labels. Through an encoder, we encode these subgraphs into rationale representations and non-rationale ones, respectively. Subsequently, a predictor is employed to exclusively leverage the extracted rationale subgraph representations for task prediction. To reduce the exploration space for identifying the correct rationale, we utilize the knowledge distillation method to align the graph representations that possess generalization capabilities learned in the *classification* module, with the rationale subgraph representations. At the end of a training iteration, we gather the non-rationale representations of all samples and employ an *environment inductor* to obtain multiple environments based on these representations. In the subsequent iteration, we introduce these environments into the *classification* module to facilitate cooperative learning between the *classification* and *rationalization* modules. Experiments over real-world benchmarks [15, 19] and various synthetic [37] datasets validate the effectiveness of our proposed C2R.

## 2 RELATED WORK

### 2.1 Graph Neural Networks

Graph Neural Networks (GNNs) have garnered significant attention and research efforts from both academia [13, 15, 39, 40] and industry [11, 36, 42]. While a plethora of methods have emerged, a substantial portion of the literature has focused on the in-distribution hypothesis [22]. This hypothesis assumed that the testing and training graph data are drawn from the same distribution. However, in real-world graph scenarios, this assumption was often violated, leading to a significant degradation in GNNs performance. Recognizing the importance of addressing this critical problem, researchers [4, 5, 7, 21, 30, 44] have increasingly turned their attention to out-of-distribution (OOD) generalization on graphs.

### 2.2 Graph Classification for Generalization

In this paper, we focused on the generalization of graph classification tasks. One intuitive approach to tackle this issue was to diversify the training data distribution by incorporating various environments during the classification process [6]. In the field of computer vision, there existed numerous related research works [8, 29, 34, 35, 45] that tailored the environment based on image data characteristics (e.g. background and color). However, it was important to note that this assumption did not directly translate to graph data, rendering this approach currently feasible.

### 2.3 Graph Rationalization for Generalization

Recent advancements in graph generalization have presented a more effective approach by exploring the concept of invariant rationale. These methods [26, 37] began by partitioning the entire graph into rationale and non-rationale subgraphs using a separator. Subsequently, through interventions in the training distribution, they identified invariant rationales under distribution shifts. DIR [37], DisC [9], CAL [32], and GREA [24] presumed that the separated non-rationale subgraphs represent the environment, and they randomly combined the rationales with other non-rationale subgraphs to create a new training distribution. DARE [43] introduced a novel disentanglement representation learning method that aimed at enhancing the distinguishability of non-rationale subgraphs. Conversely, GIL [23] utilized clustering techniques to get the local environment by grouping non-rationale subgraphs within a batch during training. However, these methods primarily relied on learning signals derived from comparing the prediction results of the rationalization method with the true labels, thereby resulting in an extensive exploration space for identifying the correct rationales. To this end, in this paper, we propose a Cooperative Classification and Rationalization (C2R) method for graph generalization, aiming to address the aforementioned problems.

## 3 PROBLEM FORMULATION

Here, we formally define the problem of graph generalization:
**General Graph Generalization.** Given the training set of $n$ instances $\mathcal{D}_G = \{(g_i, y_i)\}_i^l$, where the training distribution is $\mathcal{P}_{train}(g, y)$, the goal of graph generalization is to learn an optimal GNN $f_\theta$ that can achieve the best generalization on the data drawn from test distribution $\mathcal{P}_{test}(g, y)$ ($\mathcal{P}_{train} \neq \mathcal{P}_{test}$):

$$f_\theta^* = \arg\min_{f_\theta} \mathbb{E}_{g, y \sim \mathcal{P}_{test}} \left[ \ell \left( f_\theta(g), y \right) \right], \tag{1}$$

where $\ell(\cdot)$ denotes the loss function (e.g., the cross-entropy loss function in classification).

***Graph Classification with Environment for Generalization.***
We first suppose that the environment $E = \{\mathbf{e}_1, \mathbf{e}_2, \ldots, \mathbf{e}_k\}$ is available. Then, considering the environment $E$ and each graph $g_i = (\mathcal{V}, \mathcal{E})$ in $\mathcal{D}_G$, which consists of $|\mathcal{V}|$ nodes and $|\mathcal{E}|$ edges, the goal is to train an optimal GNN $f_\theta$ to achieve promising results in the out-of-distribution (OOD) test data:

$$f_\theta^* = \arg\min_{f_\theta} \mathbb{E}_{g, y \sim \mathcal{P}_{test}} \left[ \ell \left( f_\theta(g, E), y \right) \right]. \tag{2}$$

***Graph Rationalization for Generalization.*** Given each graph $g_i = (\mathcal{V}, \mathcal{E})$ in $\mathcal{D}_G$, the goal of graph rationalization is first to learn a mask variable $\mathbf{M} \in \mathbb{R}^{|\mathcal{V}|}$ with the separator $f_s(g_i)$ and nodes representation $\mathbf{H}_g \in \mathbb{R}^{|\mathcal{V}| \times d}$. Then, we yield the rationale subgraph representation as $\mathbf{M} \odot \mathbf{H}_{g_i}$. Finally, we learn the predictor $f_p(\mathbf{M} \odot \mathbf{H}_{g_i})$ to solve the OOD generalization problem:

$$f_s^*, f_p^* = \arg\min_{f_s, f_p} \mathbb{E}_{g, y \sim \mathcal{P}_{test}} \left[ \ell \left( f_p \left( f_s(g) \right), y \right) \right]. \tag{3}$$

## 4 COOPERATIVE CLASSIFICATION AND RATIONALIZATION FRAMEWORK

In this section, we first present the details of C2R, including both *classification* and *rationalization* modules. Then, we introduce the training and inference procedures in C2R.

### 4.1 Architecture of C2R

To learn graph classification and rationalization cooperatively to achieve the best generalization in OOD data, we propose the C2R method (Figure 2) which consists of the *classification* and *rationalization* module. Specifically, in the *classification* module, we make the graph classification (Eq.(2)) by assuming the environment is available, and further obtain the robust graph representations. Then, in the *rationalization* module, we separate the graph into the rationale subgraphs and the non-rationale ones, and transfer learned robust graph representations to rationale subgraphs. Finally, the rest non-rationale subgraphs are employed to form the multiple environments, and the environments are introduced into the *classification* module during the next training iteration.

### 4.2 The Classification Module

In the *classification* module, the graph encoder is responsible for encoding the input graph to a graph-level representation. Then, leveraging available environments, the environment-conditional generator composes multiple counterfactual samples. Finally, we employ the predictor to yield the task results based on both original and counterfactual samples, thereby addressing the OOD problem.

*4.2.1 **Graph Encoder***. In C2R, we employ any GNN structure as the graph encoder $\mathrm{GNN}_{en}(\cdot)$ (e.g., GIN [38]):

$$\mathbf{H}_{en} = \mathrm{GNN}_{en}(g), \quad \mathbf{h}_{en} = \mathrm{READOUT}\left(\mathbf{H}_{en}\right). \tag{4}$$

Among them, $\mathbf{H}_{en} \in \mathbb{R}^{|\mathcal{V}| \times d}$ is defined as the node representations, where $d$ denotes the dimensionality of the node features. The graph-level representation is denoted as $\mathbf{h}_{en} \in \mathbb{R}^d$ which is generated using a readout operator. In this paper, we employ the mean pooling as the readout operator.

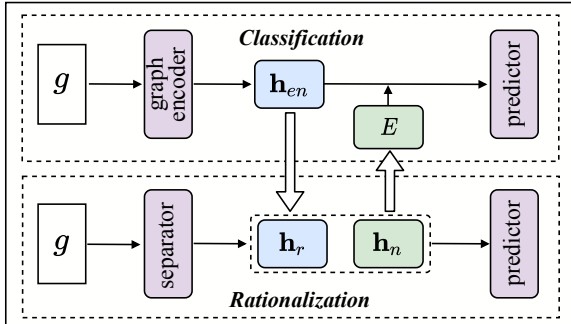

**Figure 2: Architecture of C2R, including the *classification* and *rationalization* modules.**

*4.2.2 **Environment-conditional Generator***. In this subsection, we first assume the environment set $E = \{\mathbf{e}_1, \mathbf{e}_2, \ldots, \mathbf{e}_k\}$ is available with each sample associated with a specific environment. Then, for each sample, we suppose its corresponding environment is $\mathbf{e}_m$, and we sample an environment $\mathbf{e}_j$ from $E$ randomly ($\mathbf{e}_m \neq \mathbf{e}_j$), where each $\mathbf{e}_j \in \mathbb{R}^d$ in this paper. Next, we learn an *environment-conditional generator* $\mathbb{EG}(\cdot)$ that maps an original graph level representation $\mathbf{h}_{en}$ to a novel environment distribution by conditioning on the novel environment representation $\mathbf{e}_j$:

$$\mathbf{h}_{en}^j = \mathbb{EG}\left(\mathbf{h}_{en}, \mathbf{e}_j\right), \tag{5}$$

where $\mathbf{h}_{en}^j \in \mathbb{R}^d$ and $\mathbb{EG}(\cdot)$ can be arbitrarily network architecture.

Besides, to further ensure that environment-conditional generation is effective, we design a cycle consistency constraint:

$$\mathcal{L}_{cycle} = I\left( \mathbb{EG}\left(\mathbf{h}_{en}^j, \mathbf{e}_m\right); \mathbf{h}_{en} \right), \tag{6}$$

where $\mathbb{EG}\left(\mathbf{h}_{en}^j, \mathbf{e}_m\right)$ aims to reconstruct the original $\mathbf{h}_{en}$ given the original environment $\mathbf{e}_m$. $I(;)$ denotes the mutual information which is a measure of the mutual dependence between the two variables [3, 27, 28]. By maximizing $I(;)$, we can ensure the two different representations encode the same information.

*4.2.3 **Predictor***. The predictor $\Phi(\cdot)$ yields the task results based on both the original graph representations and the counterfactual ones. It is noted that since the environment does not influence task predictions, the labels of the counterfactual samples remain unchanged. The prediction loss can be formulated as:

$$\hat{y}_{en} = \Phi(\mathbf{h}_{en}), \quad \mathcal{L}_{ori} = \mathbb{E}_{(g, y) \sim \mathcal{D}_G} \left[ \ell(\hat{y}_{en}, y) \right], \tag{7}$$

$$\hat{y}_e^j = \Phi\left(\mathbf{h}_{en}^j\right), \quad \mathcal{L}_{cou} = \mathbb{E}_{(g, y) \sim \mathcal{D}_G} \left[ \ell(\hat{y}_e^j, y) \right]. \tag{8}$$

By incorporating multiple samples, we expose the model to diverse scenarios and encourage it to generalize in various environments, contributing to improved robustness and performance.

### 4.3 The Rationalization Module

In the *rationalization* module, we employ a separator to partition the graph into two subsets: the rationale subgraphs and the non-rationale ones. Each subgraph is then encoded into its respective representations. Then, the predictor projects the rationale representations to the graph label, facilitating the classification process. Next,

to transfer learned robust graph representations to rationale subgraphs, we adopt a knowledge distillation method to align the graph representations with the rationale ones. Finally, the non-rationale subgraphs are utilized to construct multiple environments through an *environment inductor*. These environments are introduced into the *classification* module during subsequent training iteration.

*4.3.1* ***Separator in Rationalization***. The process of generating rationales in the separator consists of three steps. Initially, the separator predicts the probability distribution for selecting each node as part of the rationale:

$$\widetilde{\mathbf{M}} = \text{softmax}(W_m(\text{GNN}_m(g))), \quad (9)$$

where $W_m \in \mathbb{R}^{2 \times d}$ is a weight matrix and $\text{GNN}_m$ is an encoder that transforms each node in $g$ into a $d$-dimensional vector. Then, the separator samples the binary values (i.e., 0 or 1) from the probability distribution $\widetilde{\mathbf{M}} = \{\widetilde{m}_i\}_i^{|\mathcal{V}|}$ to form the mask variable $\mathbf{M}$. Meanwhile, to ensure differentiability during the sampling operation, we adopt the Gumbel-softmax method [16] to achieve differentiability:

$$m_j = \frac{\exp\left(\left(\log\left(\widetilde{m}_j\right) + q_j\right)/\tau\right)}{\sum_t \exp\left(\left(\log\left(\widetilde{m}_t\right) + q_t\right)/\tau\right)}, \quad (10)$$

where $\tau$ is a temperature hyperparameter, $q_j = -\log\left(-\log\left(u_j\right)\right)$, and $u_j$ is randomly sampled from a uniform distribution $U(0, 1)$.

Then, an additional GNN encoder, denoted as $\text{GNN}_g$, is employed to obtain the node representation $\mathbf{H}_g$ from the graph $g$. The rationale node representation is then defined as the element-wise product of the binary rationale mask $\mathbf{M}$ and the node representation $\mathbf{H}_g$ ($\mathbf{M} \odot \mathbf{H}_g$). Naturally, the non-rationale node representation is obtained by computing $(1 - \mathbf{M}) \odot \mathbf{H}_g$. Finally, similar to the approach described in section 4.2.1, the rationale subgraph representation $\mathbf{h}_r$ and the non-rationale subgraph representation $\mathbf{h}_n$ are computed as follows:

$$\mathbf{h}_r = \text{READOUT}(\mathbf{M} \odot \mathbf{H}_g), \quad \mathbf{h}_n = \text{READOUT}((1 - \mathbf{M}) \odot \mathbf{H}_g). \quad (11)$$

*4.3.2* ***Predictor in Rationalization***. In the *rationalization* module, we employ the predictor described in section 4.2.3 to predict the task results (i.e., the predictor in *classification* and *rationalization* share parameters) based on the rationale subgraphs:

$$\hat{y}_r = \Phi(\mathbf{h}_r), \quad \mathcal{L}_r = \mathbb{E}_{(g,y) \sim \mathcal{D}_G}[\ell(\hat{y}_r, y)]. \quad (12)$$

*4.3.3* ***Knowledge Distillation***. Since there exists a vast exploration space to compose rationales, we employ a knowledge distillation method to transfer the robust graph representation $\mathbf{h}_{en}$ learned in the *classification* module to the rationale representation $\mathbf{h}_r$. Specifically, we encourage $\mathbf{h}_r$ to match $\mathbf{h}_{en}$ for learning the generalization capability by maximizing the mutual information between $\mathbf{h}_r$ and $\mathbf{h}_{en}$:

$$\mathcal{L}_{dis} = I(\mathbf{h}_r; \mathbf{h}_{en}). \quad (13)$$

By maximizing Eq.(13), we achieve aligning the rationale representation with the robust graph representation, thereby transferring the learned knowledge.

*4.3.4* ***Environment Inductor***. After obtaining the rationale subgraph and the non-rationale subgraph, we can further infer the environment $E$. Specifically, since the non-rationale subgraph captures the correlation of the variances under different distributions, which are the environment discriminative features, we can infer potential

---

**Algorithm 1** Training process of C2R

**for** each training iteration **do**
  **for** each batch in the dataset **do**
    # In the *classification* module.
    1. Getting the graph representation $\mathbf{h}_{en}$ with Eq.(4).
    2. Generating the counterfactual samples $\mathbf{h}_{en}^j$ based on both $\mathbf{h}_{en}$ and environments $E$ with Eq.(5).
    3. Ensuring the process of counterfactual samples generation is effective with Eq.(6).
    4. Yielding task results based on both original and counterfactual samples with Eq.(7)-(8).
    # In the *rationalization* module.
    5. Separating the graph into the rationale subgraph $\mathbf{h}_r$ and non-rationale subgraph $\mathbf{h}_n$.
    6. Yielding task results based on the rationale with Eq.(12).
    7. Transferring robust graph representations $\mathbf{h}_{en}$ to the rationale $\mathbf{h}_r$ based on Eq.(13).
  **end for**
  Collecting the non-rationale subgraphs of all samples.
  Updating the environments $E$ based on Eq.(14).
**end for**

---

environments by analyzing the non-rationale subgraphs. Different from other methods [23, 24] that employ the non-rationale within each batch to infer potential environments (i.e., local environments), C2R focuses more on global environments. Therefore, we collect the non-rationale subgraphs of all samples (i.e., $\hat{h_n} = \{\mathbf{h}_n^i\}_i^l$) after each training iteration to capture a broader perspective. Among them, these non-rationale subgraphs provide insights into the overall structure and patterns across different samples.

To infer the potential global environment, we utilize the k-means clustering algorithm [14, 23] as the environment inductor:

$$E = \text{k-means}(\hat{h_n}). \quad (14)$$

After partitioning the non-rationale into multiple environments $E$, we transfer the inferred environments into the *classification* module to achieve the cooperative learning between the *classification* and *rationalization* modules.

## 4.4 Training and Inference

During training, we incorporate a sparsity constraint on the probability $\mathbf{M}$ of being selected as a rationale, following the approach proposed in [24].

$$\mathcal{L}_{sp} = \left| \frac{1}{N} \sum_{i=1}^{N} M_i - \alpha \right|, \quad (15)$$

where $\mathcal{L}_{sp}$ encourages the model to control the expected size of rationale subgraphs and $\alpha \in [0, 1]$ is the predefined sparsity level.

Finally, the overall objective of the C2R is defined as:

$$\mathcal{L} = \underbrace{\mathcal{L}_{ori} + \lambda_{cou}\mathcal{L}_{cou} - \lambda_{cycle}\mathcal{L}_{cycle}}_{classification} + \underbrace{\mathcal{L}_r + \lambda_{sp}\mathcal{L}_{sp} - \lambda_{dis}\mathcal{L}_{dis}}_{rationalization}, \quad (16)$$

where $\lambda_{cou}, \lambda_{cycle}, \lambda_{sp}$ and $\lambda_{dis}$ are adjusted hyperparameters.

Besides, in the training process, we first run the separator (section 4.3.1) and predictor serially (section 4.3.2) in the *rationalization* module to get the non-rationale of all samples. Then, we employ Eq.(14) to infer the environment $E$ which is considered as the initialization environment in the *classification* module. The process of training C2R is illustrated in Algorithm 1.

In the inference phase, both *classification* and *rationalization* modules can predict the final task results. However, the *rationalization* module can provide evidence (i.e., extracted rationales) to support the prediction results, thereby enhancing the explainability of the model. Consequently, at inference time, only $\mathbf{h}_r$ is employed to yield the task results.

## 5 EXPERIMENTS

To validate the effectiveness of the proposed C2R method, we design experiments to address the following research questions:

- **RQ1:** How effective is C2R in improving model generalization?
- **RQ2:** For the different components and hyperparameters in C2R, what are their roles and impacts on performance?
- **RQ3:** Is the cooperative training *classification* and *rationalization* strategy effective?
- **RQ4:** Can the framework of C2R help existing rationale-based methods improve the generalization?
- **RQ5:** Does C2R capture the significant rationales for predictions in the OOD dataset?

## 5.1 Datasets

To demonstrate the effectiveness of C2R, we conduct experiments on the following datasets, including synthetic and real-world datasets:

- **Synthetic Dataset.** In this study, we utilize the Spurious-Motif dataset [37, 41] as our synthetic dataset for predicting motif types. Each graph in the Spurious-Motif dataset comprises two subgraphs: the motif subgraph denoted as $R$ and the base subgraph denoted as $B$. The motif subgraph serves as the rationale for motif type prediction and consists of three types: *Cycle*, *House*, and *Crane* (represented as $R = 0, 1, 2$ respectively). On the other hand, the base subgraph varies with the motif type and can be viewed as the non-rationale. It consists of three types: *Tree*, *Ladder*, and *Wheel* (denoted as $B = 0, 1, 2$ respectively). An example of the Spurious-Motif dataset, specifically the *House-Tree* combination, is illustrated in Figure 1.

  To demonstrate that C2R can achieve promising experimental results on the OOD data, we manually construct the Spurious-Motif dataset with different data distributions. Specifically, in this construction process, we sample the motif subgraph uniformly and select the base subgraph based on the following distribution:

$$P(E) = \begin{cases} bias, & \text{if } B = R \\ \frac{1-bias}{2}, & \text{if } B \neq R \end{cases}, \tag{17}$$

  where the parameter *bias* controls the extent of data distributions (the higher the *bias*, the more significant the spurious correlation in the data.) In this study, we consider three Spurious-Motif datasets with *bias* values of 0.5, 0.7, and 0.9. Additionally, for fair evaluation, a de-biased (balanced) dataset is created for the test set by setting $bias = \frac{1}{3}$.

**Table 1: Statistics of Synthetic and Real-world Datasets.**

| Dataset | Train/Val/Test | Classes | Avg. Nodes | Avg. Edges |
|---|---|---|---|---|
| Spurious-Motif (bias=0.5) | 3,000/3,000/6,000 | 3 | 29.6 | 42.0 |
| Spurious-Motif (bias=0.7) | 3,000/3,000/6,000 | 3 | 30.8 | 45.9 |
| Spurious-Motif (bias=0.9) | 3,000/3,000/6,000 | 3 | 29.4 | 42.5 |
| MolHIV | 32,901/4,113/4,113 | 2 | 25.5 | 27.5 |
| MolToxCast | 6,860/858/858 | 617 | 18.8 | 19.3 |
| MolBBBP | 1,631/204/204 | 2 | 24.1 | 26.0 |
| MolSIDER | 1,141/143/143 | 27 | 33.6 | 35.4 |
| MNIST-75sp | 5,000/1,000/1,000 | 10 | 66.8 | 600.2 |

- **MNIST-75sp** [19]. In this dataset, each image from the MNIST [20] dataset is transformed into a superpixel graph, with a maximum of 75 nodes per graph. Besides, to simulate the scenario where the model faces the OOD data during testing, random noises are introduced to the node features of the superpixel graphs.
- **OGB.** For real-world datasets, we utilize the Open Graph Benchmark (OGB) [15]. Specifically, we focus on the OGB-Mol datasets available within OGB, including MolHIV, MolToxCast, MolBBBP, and MolSIDER, which provide diverse molecular properties for analysis and prediction. To ensure a consistent and standardized evaluation, we adopt the default scaffold splitting method employed by OGB. This method partitions the datasets into training, validation, and test sets based on molecular scaffolds.

Details of dataset statistics are shown in Table 1.

## 5.2 Comparison Methods

In this section, we first present several rationale-based methods for graph generalization.

- **DIR** [37] introduces a new strategy for discovering invariant rationale (DIR) to compose rationales. DIR conducts interventions on the training distribution to create multiple interventional distributions, enhancing the generalizability of DIR.
- **DisC** [9] designs a general disentangling framework to learn the causal substructure and bias substructure and synthesizes the counterfactual training samples to further de-correlate causal and bias variables.
- **GREA** [24] is another method that generates counterfactual samples using the bias substructure. However, unlike other approaches (e.g. DisC), there exists no disentanglement operation in GERA to ensure the bias can be separated from the original input.
- **CAL** [32] proposes the Causal Attention Learning (CAL) strategy, which discovers the causal rationales and mitigates the confounding effect of shortcuts to achieve high generalization.
- **GSAT** [26] proposes a method that introduces stochasticity to block label-irrelevant information and selectively identifies label-relevant subgraphs. This selection process is guided by the information bottleneck principle [1, 33].
- **GIL** [23] learns generalized graph representations under local environments shift, where the local environments are inferred by clustering the non-rationales of a batch.
- **DARE** [43] introduces a self-guided method with the disentanglement operation to encapsulate more information from the input to extract rationales.

**Table 2: Performance on the Synthetic Dataset and Real-world Datasets.**

| | | Spurious-Motif (ACC) | | | OGB (AUC) | | | | MNIST-75sp |
| --- | --- | --- | --- | --- | --- | --- | --- | --- | --- |
| | | bias=0.5 | bias=0.7 | bias=0.9 | MolHIV | MolToxCast | MolBBBP | MolSIDER | (ACC) |
| GIN is the backbone | GIN | $0.3950 \pm 0.0471$ | $0.3872 \pm 0.0531$ | $0.3768 \pm 0.0447$ | $0.7447 \pm 0.0293$ | $0.6521 \pm 0.0172$ | $0.6584 \pm 0.0224$ | $0.5977 \pm 0.0176$ | $0.1201 \pm 0.0042$ |
| | DIR | $0.4444 \pm 0.0621$ | $0.4891 \pm 0.0761$ | $0.4131 \pm 0.0652$ | $0.6303 \pm 0.0607$ | $0.5451 \pm 0.0092$ | $0.6460 \pm 0.0139$ | $0.4989 \pm 0.0115$ | $0.1893 \pm 0.0458$ |
| | DisC | $0.4585 \pm 0.0660$ | $0.4885 \pm 0.1154$ | $0.3859 \pm 0.0400$ | $0.7731 \pm 0.0101$ | $0.6662 \pm 0.0089$ | $0.6963 \pm 0.0206$ | $0.5846 \pm 0.0169$ | $0.1262 \pm 0.0113$ |
| | GERA | $0.4251 \pm 0.0458$ | $0.5331 \pm 0.1509$ | $0.4568 \pm 0.0779$ | $0.7714 \pm 0.0153$ | $0.6694 \pm 0.0043$ | $0.6953 \pm 0.0229$ | $0.5864 \pm 0.0052$ | $0.1172 \pm 0.0021$ |
| | CAL | $0.4734 \pm 0.0681$ | $0.5541 \pm 0.0323$ | $0.4474 \pm 0.0128$ | $0.7339 \pm 0.0077$ | $0.6476 \pm 0.0066$ | $0.6582 \pm 0.0397$ | $0.5965 \pm 0.0116$ | $0.1258 \pm 0.0123$ |
| | GSAT | $0.4517 \pm 0.0422$ | $0.5567 \pm 0.0458$ | $0.4732 \pm 0.0367$ | $0.7524 \pm 0.0166$ | $0.6174 \pm 0.0069$ | $0.6722 \pm 0.0197$ | $0.6041 \pm 0.0096$ | $0.2381 \pm 0.0186$ |
| | DARE | $0.4843 \pm 0.1080$ | $0.4002 \pm 0.0404$ | $0.4331 \pm 0.0631$ | $0.7836 \pm 0.0015$ | $0.6677 \pm 0.0058$ | $0.6820 \pm 0.0246$ | $0.5921 \pm 0.0260$ | $0.1201 \pm 0.0042$ |
| | GIL | $0.5013 \pm 0.0973$ | $0.5731 \pm 0.0722$ | $0.5501 \pm 0.0834$ | $0.7868 \pm 0.0174$ | $0.6690 \pm 0.0048$ | $0.6901 \pm 0.0569$ | $0.6083 \pm 0.0051$ | $0.2108 \pm 0.0094$ |
| | C2R | $\mathbf{0.5203 \pm 0.1437}$ | $\mathbf{0.5913 \pm 0.0413}$ | $\mathbf{0.5601 \pm 0.0979}$ | $\mathbf{0.7919 \pm 0.0006}$ | $\mathbf{0.6709 \pm 0.0052}$ | $\mathbf{0.6999 \pm 0.0122}$ | $\mathbf{0.6131 \pm 0.0117}$ | $\mathbf{0.2433 \pm 0.0311}$ |
| GCN is the backbone | GCN | $0.4091 \pm 0.0398$ | $0.3772 \pm 0.0763$ | $0.3566 \pm 0.0323$ | $0.7128 \pm 0.0188$ | $0.6497 \pm 0.0114$ | $0.6665 \pm 0.0242$ | $0.6108 \pm 0.0075$ | $0.1195 \pm 0.0149$ |
| | DIR | $0.4281 \pm 0.0520$ | $0.4471 \pm 0.0312$ | $0.4588 \pm 0.0840$ | $0.4258 \pm 0.1084$ | $0.5077 \pm 0.0094$ | $0.5069 \pm 0.1099$ | $0.5224 \pm 0.0243$ | $0.1798 \pm 0.0328$ |
| | DisC | $0.4698 \pm 0.0408$ | $0.4312 \pm 0.0358$ | $0.4713 \pm 0.1390$ | $0.7791 \pm 0.0137$ | $0.6626 \pm 0.0055$ | $\mathbf{0.7061 \pm 0.0105}$ | $0.6110 \pm 0.0091$ | $0.1262 \pm 0.0113$ |
| | GERA | $0.4687 \pm 0.0855$ | $0.5467 \pm 0.0742$ | $0.4651 \pm 0.0881$ | $0.7816 \pm 0.0079$ | $0.6622 \pm 0.0045$ | $0.6970 \pm 0.0089$ | $0.6133 \pm 0.0239$ | $0.1160 \pm 0.0140$ |
| | CAL | $0.4245 \pm 0.0152$ | $0.4355 \pm 0.0278$ | $0.3654 \pm 0.0064$ | $0.7501 \pm 0.0094$ | $0.6006 \pm 0.0031$ | $0.6635 \pm 0.0257$ | $0.5559 \pm 0.0151$ | $0.1043 \pm 0.0080$ |
| | GSAT | $0.3630 \pm 0.0444$ | $0.3601 \pm 0.0419$ | $0.3929 \pm 0.0289$ | $0.7598 \pm 0.0085$ | $0.6124 \pm 0.0082$ | $0.6437 \pm 0.0082$ | $0.6179 \pm 0.0041$ | $0.2549 \pm 0.0123$ |
| | DARE | $0.4609 \pm 0.0648$ | $0.5035 \pm 0.0247$ | $0.4494 \pm 0.0526$ | $0.7523 \pm 0.0041$ | $0.6618 \pm 0.0065$ | $0.6823 \pm 0.0068$ | $0.6192 \pm 0.0079$ | $0.1106 \pm 0.0086$ |
| | GIL | $0.4997 \pm 0.0485$ | $0.5580 \pm 0.0481$ | $0.5086 \pm 0.0874$ | $0.7808 \pm 0.0093$ | $0.6530 \pm 0.0098$ | $0.6808 \pm 0.0083$ | $0.6177 \pm 0.0045$ | $0.2290 \pm 0.0469$ |
| | C2R | $\mathbf{0.5161 \pm 0.0199}$ | $\mathbf{0.5835 \pm 0.0972}$ | $\mathbf{0.6203 \pm 0.0304}$ | $\mathbf{0.7899 \pm 0.0088}$ | $\mathbf{0.6671 \pm 0.0040}$ | $0.6916 \pm 0.0195$ | $\mathbf{0.6256 \pm 0.0106}$ | $\mathbf{0.2591 \pm 0.0381}$ |

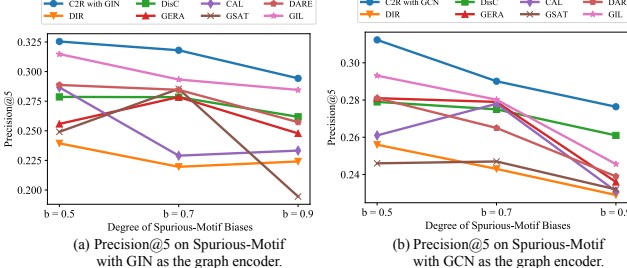

(a) Precision@5 on Spurious-Motif with GIN as the graph encoder.

(b) Precision@5 on Spurious-Motif with GCN as the graph encoder.

**Figure 3: Results of Precision@5 between extracted rationales and the ground-truth rationales on Spurious-Motif.**

Additionally, we conduct a comparative analysis by considering several conventional GNN architectures for classification, including GCN [18] and GIN [38]. Meanwhile, we employ both GCN and GIN as the backbone of C2R and other baselines.

## 5.3 Experimental Setup

In all experimental settings, the values of the hyperparameters $\lambda_{cou}$, $\lambda_{cycle}$, $\lambda_{sp}$, and $\lambda_{dis}$ are uniformly set to 1.0, 0.01, 0.01, and 1.0, respectively. The hidden dimensionality $d$ is specifically configured as 32 for the Spurious-Motif dataset, 64 for the MNIST-75sp dataset, and 128 for the OGB dataset. During the training process, we employ the Adam optimizer [17] with a learning rate initialized as 1e-2 for the Spurious-Motif and MNIST-75sp datasets, and 1e-3 for the OGB dataset. We set the predefined sparsity $\alpha$ as 0.1 for MolHIV, 0.5 for MolSIDER, MolToxCast and MolBBBP, and 0.4 for other datasets. The number of clusters $k$ is 3 for Spurious-Motif and 10 for other datasets. In this paper, we employ a single Multilayer Perceptron (MLP) as $\mathbb{E}\mathbb{G}(\cdot)$ that takes $[\mathbf{h}_{en}; \mathbf{e}]$ as the input, where ";" represents the concatenation operation. Besides, to achieve MI maximization, we employ the InfoNCE method proposed by [28].

During the evaluation phase, we employ the ACC metric to evaluate the task prediction performance for the Spurious-Motif and MNIST-75sp datasets, and AUC for OGB. Moreover, as the Spurious-Motif includes ground-truth rationales, we evaluate the performance of the extracted rationales using precision metrics, specifically Precision@5. This metric measures the precision of the top 5 extracted rationales compared to the ground truth, providing insights into the accuracy of the rationale extraction process. All methods are trained with five different random seeds on a single A100 GPU. The reported test performance includes the mean results and standard deviations obtained from the epoch that attains the highest validation prediction performance.

## 5.4 Performance on both Synthetic and Real-world Datasets (RQ1).

To verify the effectiveness of C2R on graph generalization, we first compare the performance of C2R and other baselines on the task prediction. Specifically, as shown in Table 2, we can observe that both GIN and GCN perform poorly in prediction on OOD data, illustrating the necessity of exploring how to enhance the generalization ability of GNNs. DIR, DisC, CAL and GREA all assume that the separated non-rationale is the environment and promote its generalization ability under the environment shifts. Although such methods can achieve promising results, they are still lower than C2R. This observation suggests that employing coarse-grained environment inference methods that treat each non-rationale as a distinct environment is suboptimal. Conversely, the environment derived from inductive clustering of all non-rationales proves to be more representative and facilitates model effectiveness.

Meanwhile, GIL also performs better than DIR and DisC. Among them, GIL attains the local environment by clustering non-rationales within a batch, highlighting the necessity of leveraging non-rationale clustering to obtain effective environments. Nonetheless, GIL's performance still lags behind that of C2R, suggesting that the global environment inferred from all non-rationale subgraphs is more impactful than the local environment. Finally, both GSAT and DARE

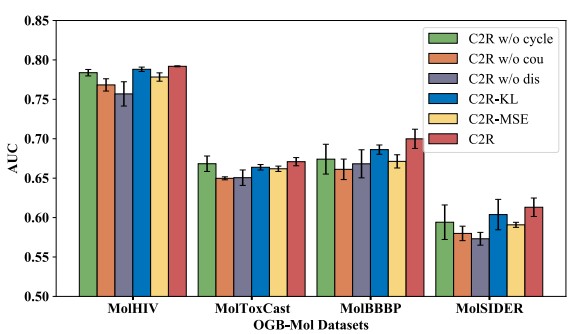

Figure 4: Ablation study and Hyperparameter Sensitivity Analysis of C2R which is implemented with GIN over OGB.

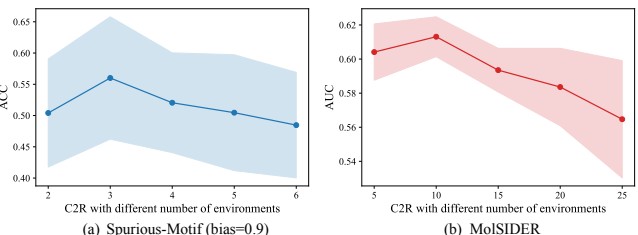

(a) Spurious-Motif (bias=0.9)  (b) MolSIDER

Figure 5: Hyperparameter Sensitivity Analysis of the number of inductive environments $k$.

fully exploit the potential of rationalization itself and achieve commendable results on OOD generalization tasks, where these methods can be considered as self-guided approaches. However, their results are still lower than the C2R using external guidance (knowledge distillation). This comparison strongly implies that C2R can be more effective in reducing the exploration space of composing rationale, thereby enabling superior performance.

Besides, to further analyze whether C2R realistically captures the invariant rationale for graph generalization, we conduct experiments in Spurious-Motif which contains the real rationale. Specifically, we present the Precision@5 values which measure the precision of the top 5 extracted rationales compared to the gold rationales in Figure 3. The results reveal that regardless of the degree of bias in the Spurious-Motif dataset (ranging from 0.5 to 0.9), the rationales extracted by C2R consistently exhibit higher accuracy compared to the baseline methods. This finding underscores the effectiveness of cooperative classification and rationalization training employed in C2R.

## 5.5 Ablation Study and Hyperparameter Sensitivity Analysis (RQ2).

**Ablation Study.** In this section, we first validate the effectiveness of each component in C2R through ablation experiments, focusing on three aspects:
(*i*). We remove the cycle consistency constraint (i.e., Eq.(6)), and we name this variant as C2R w/o cycle.
(*ii*). We investigate the impact of counterfactual samples on prediction by eliminating their usage (Eq.(8)). This variant is referred to as C2R w/o cou.
(*iii*). We remove the process of knowledge distillation, such that C2R degenerates into a simple classification and rationalization multi-task learning method and is named C2R w/o dis.

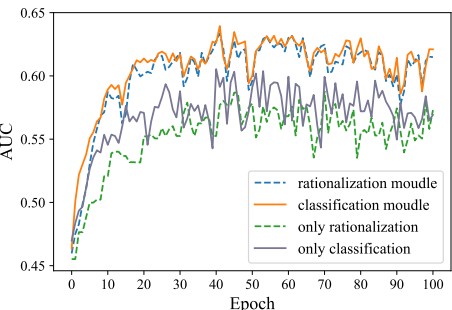

Figure 6: Training process of C2R on MolSIDER.

We conduct experiments on the OGB dataset with GIN as the backbone. Specifically, as shown in Figure 4, we observe that:
(*i*). The performance of C2R w/o cycle is inferior to that of C2R, indicating that the cycle consistency constraint plays a crucial role in enhancing the effectiveness of the environmental condition generator. This constraint ensures that the generated counterfactual samples adhere to the predefined environmental distribution. Nonetheless, C2R w/o cycle still outperforms most baselines, demonstrating the effectiveness of the core framework based on cooperative classification and rationalization.
(*ii*). The removal of counterfactual samples in C2R (C2R w/o cou) leads to a significant decrease. This decline can be attributed to the fact that counterfactual samples contribute to diversified data distributions, which enhance the model's generalization ability.
(*iii*). C2R w/o dis exhibits poor performance due to the absence of the knowledge distillation process. Without knowledge distillation, the robust graph representations learned by the classification module cannot be transferred effectively to the rationalization module. Consequently, it becomes challenging to improve the generalization ability of the rationalization module solely through the multi-task learning framework between the classification and rationalization.

**Hyperparameter Sensitivity Analysis.** Besides, we investigate the sensitivity of hyper-parameters of C2R, including the number of inductive environments $k$ and the alignment method used for the knowledge distillation process. Specifically, since the number of environments is important for the ability of the classification module to generate robust graph representations, we conduct experiments on the Spurious-Motif (bias=0.9) and MolSIDER datasets with GIN as the backbone. As shown in Figure 5(a), for the Spurious-Motif dataset, the optimal number of environments is 3, which is consistent with the true number of environments in Spurious-Motif (i.e., $|B|$ = 3 in section 5.1). For the MolSIDER dataset (Figure 5(b)), the optimal value of $k$ is 10. Moreover, we find when the number of environments is relatively high ($k \geq 20$), the model performs mediocrely, indicating that too many environments do not yield a significant gain on the model's effectiveness.

To align the robust graph representations with the rationale representations, we employ the method of maximizing Mutual Information (MI) (Eq.(13)). To validate the effectiveness of MI maximization, we compare it with two alternative methods: (1) Minimizing the Kullback-Leibler (KL) divergence between $\mathbf{h}_r$ and $\mathbf{h}_{en}$ (C2R-KL). (2) Minimizing the Mean Squared Error (MSE) between $\mathbf{h}_r$ and $\mathbf{h}_{en}$ (C2R-MSE). Figure 4 illustrates the impact of these replacements, where we can observe that using the method of MI maximization can effectively align $\mathbf{h}_r$ and $\mathbf{h}_{en}$.

**Table 3: Structural Generalizability of C2R. Each rationalization method in C2R is highlighted with a gray background**

| | | MolHIV | MolToxCast | MolBBBP | MolSIDER |
|---|---|---|---|---|---|
| GIN is the backbone | DisC | 0.7731 | 0.6662 | 0.6963 | 0.5846 |
| | DisC+C2R | 0.7959 (↑ 2.28%) | 0.6798 (↑ 1.36%) | 0.7031 (↑ 0.68%) | 0.6001 (↑ 1.55%) |
| | GERA | 0.7714 | 0.6694 | 0.6953 | 0.5864 |
| | GERA+C2R | 0.7993 (↑ 2.79%) | 0.6781 (↑ 0.87%) | 0.7093 (↑ 1.40%) | 0.5992 (↑ 1.28%) |
| | GSAT | 0.7524 | 0.6174 | 0.6722 | 0.6041 |
| | GSAT+C2R | 0.7793 (↑ 2.69%) | 0.6419 (↑ 2.45%) | 0.6983 (↑ 2.61%) | 0.6139 (↑ 0.98%) |
| | DARE | 0.7836 | 0.6677 | 0.6820 | 0.5921 |
| | DARE+C2R | 0.7982 (↑ 1.46%) | 0.6801 (↑ 1.24%) | 0.6965 (↑ 1.45%) | 0.6191 (↑ 2.70%) |
| GCN is the backbone | DisC | 0.7791 | 0.6626 | 0.7061 | 0.6110 |
| | DisC+C2R | 0.7902 (↑ 1.11%) | 0.6772 (↑ 1.46%) | 0.7192 (↑ 1.31%) | 0.6293 (↑ 1.83%) |
| | GERA | 0.7816 | 0.6622 | 0.6970 | 0.6133 |
| | GERA+C2R | 0.7987 (↑ 1.71%) | 0.6739 (↑ 1.17%) | 0.7034 (↑ 0.64%) | 0.6209 (↑ 0.76%) |
| | GSAT | 0.7598 | 0.6124 | 0.6437 | 0.6179 |
| | GSAT+C2R | 0.7787 (↑ 1.89%) | 0.6198 (↑ 0.74%) | 0.6683 (↑ 2.46%) | 0.6201 (↑ 0.22%) |
| | DARE | 0.7523 | 0.6618 | 0.6823 | 0.6192 |
| | DARE+C2R | 0.7801 (↑ 2.78%) | 0.6721 (↑ 1.03%) | 0.7094 (↑ 2.71%) | 0.6203 (↑ 0.11%) |

## 5.6 Training Process of C2R (RQ3).

In this section, we investigate the training process of C2R to analyze the effectiveness of our cooperative training strategy. Specifically, we make experiments on MolSIDER with the GIN backbone. Figure 6 showcases the changes in the *classification* and *rationalization* modules' AUC on MolSIDER test set over training epochs, where both modules can yield prediction results. Besides, we also compare C2R with the vanilla classification and rationalization method, both of which encounter challenges in solving the OOD problem.

From the figure, it is evident that the AUC of both the *classification* and *rationalization* modules in C2R consistently surpasses that of the vanilla classification and rationalization method throughout the training process. This observation emphasizes the necessity of cooperative training for the *classification* and *rationalization* modules. Additionally, we note that in the initial stages of training, the AUC of the *classification* module exceeds that of the *rationalization* module. This discrepancy may be attributed to the *rationalization* module initially capturing insufficient rationale to support accurate task predictions. However, as C2R undergoes further cooperative training, the gap between the *classification* and *rationalization* modules diminishes. This trend illustrates the effectiveness of our cooperative strategy. Finally, considering the relatively small difference in AUC between the *classification* and *rationalization* modules, and the *rationalization* module's ability to extract rationale as evidence for prediction results, we employ the *rationalization* module to generate task results for evaluation, as described in section 4.4.

## 5.7 Structural Generalizability of C2R (RQ4).

In C2R, we employ a classical rationale extraction framework as the rationalization module. However, an interesting question arises: Can our proposed C2R help other more advanced rationale-based methods to improve generalizability? To investigate this, we replace the rationalization module in C2R with advanced methods such as DisC, GREA, GSAT, and DARE, respectively. We then conduct experiments on OGB data to evaluate the performance. Table 3 presents the experimental results, demonstrating that C2R consistently improves the effectiveness of all rationale-based baselines.

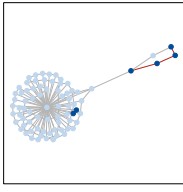 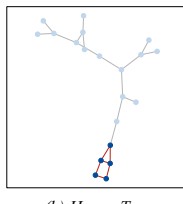 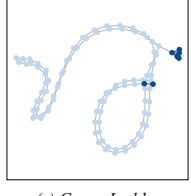

*(a) Cycle-Wheel*  *(b) House-Tree*  *(c) Crane-Ladder*

**Figure 7: Visualization of C2R rationale subgraphs.**

This finding indicates that our proposed cooperative-based C2R framework is capable of enhancing the generalizability of different rationale-based methods. These experimental results validate the effectiveness of the C2R framework and highlight its potential as a valuable framework for enhancing the generalizability and performance of different rationale-based methods.

## 5.8 Case Study (RQ5).

In this section, we provide visualizations of C2R on the test set, which is trained in Spurious-Motif (bias=0.9) and GIN serves as the backbone. Figure 7 illustrates several rationale subgraphs extracted by C2R. Each graph in the figure represents a motif type, such as *Cycle*, *House*, and *Crane*, combined with a base, such as *Tree*, *Wheel*, and *Ladder*. The navy blue nodes highlighted in the graph indicate the selected rationale nodes. Meanwhile, we assume that if there is an edge between the two identified nodes, we will visualize this edge as the red lines.

By examining the figure, we can observe that C2R successfully extracts accurate rationales for prediction. The visualized rationales demonstrate the model's ability to identify important nodes within the graph, providing meaningful insights into the decision-making process of C2R. These visualizations highlight the effectiveness of the C2R approach in extracting accurate and informative rationales from graph data, thereby enhancing the model's explainability and overall performance. This advantage again supports the reason why we employ the *rationalization* module instead of the *classification* module in our inference process.

## 6 CONCLUSION

In this paper, we proposed a Cooperative Classification and Rationalization (C2R) method for graph generalization, consisting of the *classification* and *rationalization* modules. To be specific, in the *classification* module, we first assumed that multiple environments are available. Then, we created counterfactual samples with an environment-conditional generator to enrich the training distributions. By predicting the task results based on both original and counterfactual samples, we could get robust graph representations. Besides, in the *rationalization* module, we employed a separator to partition the graph into rationale and non-rationale subgraphs. We then transferred the robust graph representations to the rationale with a knowledge distillation method. At the end of each training iteration, we gathered non-rationales of all samples and adopted the environment inductor to infer the global environments. Finally, the environments were transferred to the *classification* module to achieve cooperative training. Experimental results on five real-world datasets and three synthetic datasets have clearly demonstrated the effectiveness of our proposed method.

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
