# OpenReview forum: "Cooperative Classification and Rationalization for Graph Generalization"
_ACM.org/TheWebConf/2024/Conference — TheWebConf24_

### Official Review · Reviewer_eB73 · 2023-11-22

**Novelty:** 5
**Technical Quality:** 6

**Review:**

1)	Quality and Clarity
The paper proposed a Cooperative Classification and Rationalization(C2R) method, combining classification and rationalization modules for graph generalization. This dual-module approach is an original idea in solving the out-of-distribution(OOD) data problem. The architecture of C2R, including the classification and rationalization modules is well-written, offering clarity on how the model works. The ablation study and hyperparameter sensitivity analysis show some important components contributing to the robustness of the proposed C2R method.
2)	Originality
The C2R model presents a unique approach by integrating classification with rationalization using the knowledge distillation methods. However, the use of GNNs for graph classification and the concept of rationalization, are already well-explored in existing literature. The paper could have better highlighted the unique aspects and strengths of its approach compared with other graph rationalization methods, such as DIR, GIL, RGCL and GREA.
3)	Significance
The paper tackles an important issue of OOD generalization in graph neural networks, which has broad implications for the field. The C2R model effectively captured the invariant rationale for graph generalization and outperformed several baseline models. Structural generalizability tests indicated that the C2R framework could improve the generalizability of other rationale-based methods, demonstrating its potential to enhance the performance and generalizability of graph neural networks in various applications.

**Questions:**

1)	Related work: Could the author better highlight the unique aspects and strengths of the C2R method over other graph rationalization methods, such as DIR, GIL, RGCL, and GREA?
2)	Methodology: During the inference, how does the rationalization module learn from the classification module, and can it override the classification module if there is a conflict?
3)	Dataset: How well do the synthetic datasets represent the characteristics and complexity of real-world graph data?
4)	Experiment: In the hyperparameter sensitivity analysis, the author mentioned, “When the number of environments is relatively high, the model performs mediocrely.”, is the C2R framework robust to handle graphs with heterogeneous data?
5)	Spelling errors: In Figure 6, the legend of “rationalization moudle” and “classification moudle” should be “rationalization module” and “classification module”.

**Reviewer Confidence:**

3: The reviewer is confident but not certain that the evaluation is correct

**Scope:**

3: The work is somewhat relevant to the Web and to the track, and is of narrow interest to a sub-community

---

### Official Review · Reviewer_CRbG · 2023-11-24

**Novelty:** 5
**Technical Quality:** 5

**Review:**

The authors  propose a Cooperative Classification and Rationalization (C2R) method for graph generalization.
The method  consis of  classification and rationalization modules.
In classification module, the authors use  multiple  and  create counterfactual samples  to improve the training distributions.

Further, they propose a rationalization module,  to partition the graph into rationale and non-rationale subgraphs.


The authors evaluate their approach on real world datasets. The results show the effectives of the proposed approach, especially on Molecular datasets, and MNIST dataset.


Cons:
Code is not shared. There is a link but it is empty.

**Questions:**

1. How sensitive is the approach to the pooling operator ( mean is currently used). What is the impact of using sum pool. Was there a reason for choosing mean pool.

2. Code link is empty. Any specific reason of not sharing code while giving the link.

3. How many runs were conducted for reporting mean and std dev?

**Reviewer Confidence:**

2: The reviewer is willing to defend the evaluation, but it is likely that the reviewer did not understand parts of the paper

**Scope:**

3: The work is somewhat relevant to the Web and to the track, and is of narrow interest to a sub-community

---

### Official Review · Reviewer_kNzz · 2023-11-27

**Novelty:** 5
**Technical Quality:** 4

**Review:**

This paper presents a novel framework called C2R  that combines classification and rationalization techniques for graph data analysis.  The authors propose a cooperative learning approach that synergistically trains the classification and rationalization modules to enhance the performance and interpretability of graph-based models.  The paper introduces several notable contributions.  Firstly, it presents a cooperative learning framework that simultaneously trains the classification and rationalization modules.  This approach enables the model to learn discriminative features for accurate classification and interpretable rationales for explaining predictions.  Secondly, the paper proposes a counterfactual sampling method to generate diverse rationales, which aids in capturing different perspectives and enhancing the model's robustness.  Thirdly, the authors employ a knowledge distillation technique to transfer learned knowledge from the classification module to the rationalization module, thereby improving the generalization capability of the model.
The paper is well-articulated and provides a clear explanation of the proposed framework.  The authors offer comprehensive descriptions of the model's architecture and training process, facilitating easy understanding and reproducibility of the experiments.  The experimental results presented in the paper demonstrate the effectiveness of the proposed framework, showcasing improvements in both classification performance and rationalization quality when compared to baseline methods.
One of the strengths of this paper lies in its integration of classification and rationalization modules, enabling a more comprehensive analysis of graph data.  Additionally, the counterfactual sampling method and knowledge distillation technique contribute significantly to the interpretability and generalization of the proposed model.

**Questions:**

1.	Could you please some comparision between the proposed method with some OOD generalization methods?
2.	How does the counterfactual sampling method used in the classification module ensure the generation of diverse counterfactual samples while maintaining unaltered labels for the data?
3.	Could you provide a more in-depth explanation of how the knowledge distillation method aligns the graph representations with the rationale subgraph representations?  How does this alignment contribute to reducing the exploration space for identifying the correct rationales?
4.	Are there specific assumptions or limitations of the C2R method that should be taken into consideration when applying it to real-world scenarios?
5.	How does the rationalization module precisely identify and extract subsets of rationale subgraphs?  Can you provide a detailed explanation of the process involved in encoding these subgraphs into rationale and non-rationale representations?

**Reviewer Confidence:**

3: The reviewer is confident but not certain that the evaluation is correct

**Scope:**

4: The work is relevant to the Web and to the track, and is of broad interest to the community

---

### Official Review · Reviewer_td8U · 2023-12-08

**Novelty:** 4
**Technical Quality:** 4

**Review:**

**Evaluation of Cooperative Classification and Rationalization for Graph Generalization**

The Cooperative Classification and Rationalization (C2R) method for graph generalization presents an innovative approach to addressing the challenges of generalizing effectively with out-of-distribution (OOD) data in graph classification tasks. The method combines classification and rationalization modules to enhance the accuracy of predictions and extract invariant rationales for predictions. Here is an evaluation of the quality, clarity, originality, and significance of this work, along with a list of its pros and cons.

**Quality:**
The quality of the C2R method is commendable. The authors have provided a thorough and well-structured explanation of the method, including the classification and rationalization modules. The method is supported by rigorous experimental evaluations on both real-world and synthetic datasets, demonstrating its effectiveness in improving graph generalization. The use of metrics such as ACC, AUC, and Precision@5 for evaluation adds to the robustness of the study. Additionally, the method's ability to address the limitations of existing approaches in graph generalization is a testament to its quality.

**Clarity:**
The paper is well-written and effectively communicates the concepts and methodologies involved in the C2R method. The authors provide clear explanations of the rationale behind each component of the method, making it accessible to readers with varying levels of expertise in graph neural networks and machine learning. The use of visualizations, such as Figure 7, further enhances the clarity of the method's application and results.

**Originality:**
The C2R method demonstrates a high degree of originality in its approach to graph generalization. By integrating the classification and rationalization modules, the method offers a novel solution to the challenge of OOD data in graph classification tasks. The use of counterfactual samples, environment-conditional generators, and knowledge distillation methods contributes to the originality of the approach. Furthermore, the method's emphasis on extracting accurate and informative rationales from graph data sets it apart from traditional graph classification techniques.

**Significance:**
The significance of the C2R method lies in its potential to advance the field of graph neural networks and machine learning. By addressing the limitations of existing approaches and demonstrating superior performance in graph generalization tasks, C2R has the potential to impact various domains, including bioinformatics, social network analysis, and recommendation systems. The method's ability to enhance model explainability through rationale extraction further adds to its significance in the context of interpretable machine learning.

**Pros:**
- Rigorous experimental evaluations on real-world and synthetic datasets
- Clear and accessible explanations of the method's components and applications
- Original approach to addressing the challenges of OOD data in graph classification
- Potential to enhance model explainability through rationale extraction
- Demonstrated superiority over existing approaches in graph generalization tasks

**Cons:**
- Limited discussion on potential limitations or challenges in the implementation of the C2R method
- The paper could benefit from a more extensive comparison with state-of-the-art methods in graph generalization
- Further exploration of the scalability and computational efficiency of the method may be beneficial for practical applications

In conclusion, the Cooperative Classification and Rationalization (C2R) method represents a high-quality, original, and significant contribution to the field of graph generalization. Its potential to improve model accuracy, explainability, and generalization in graph classification tasks makes it a valuable addition to the existing body of research in machine learning and graph neural networks.

**Questions:**

Can you provide more insights into the practical implications of the Cooperative Classification and Rationalization (C2R) method in real-world applications, particularly in domains where explainability and generalization are crucial?

How does the C2R method handle scalability when dealing with large-scale graph datasets, and are there any potential limitations or trade-offs in terms of computational efficiency?

Could you elaborate on the potential challenges or limitations of the C2R method when applied to highly dynamic or evolving environments, and how the model adapts to changes in the data distribution over time?

In the context of the rationalization module, how does the C2R method address the trade-off between extracting accurate rationales and maintaining model interpretability, especially in complex graph structures?

Considering the experimental results, could you discuss the robustness of the C2R method in scenarios with noisy or incomplete graph data, and how the model's performance may be affected under such conditions?

How does the C2R method handle adversarial attacks or perturbations in graph data, and what measures are in place to ensure the model's resilience in such scenarios?

Can you provide insights into the interpretability of the rationale subgraphs extracted by the C2R method, particularly in terms of providing meaningful insights for domain experts to understand the decision-making process of the model?

Given the ablation study results, could you discuss the potential implications of removing specific components of the C2R method on the model's overall performance and generalization capabilities, and how these findings may impact the practical deployment of the model?

What are the key considerations and potential avenues for future research in further enhancing the Cooperative Classification and Rationalization (C2R) method, particularly in addressing emerging challenges in graph-based classification and explainable AI?

How does the C2R method compare to existing state-of-the-art approaches in terms of model explainability, generalization, and scalability, and what are the unique advantages that set C2R apart from other methods in the field?

**Reviewer Confidence:**

3: The reviewer is confident but not certain that the evaluation is correct

**Scope:**

3: The work is somewhat relevant to the Web and to the track, and is of narrow interest to a sub-community

---

### Decision · Program_Chairs · 2024-01-22

**Decision:**

Accept

**Comment:**

The research question is of broad interest to the community. Most reviewers think that the paper is clearly written and the approach is novel. More experiments are also conducted during rebuttal phase to address reviewers' questions.